# Phosphorus supply and floodplain design govern phosphorus reduction capacity in remediated agricultural streams

Lukas Hallberg[1], Faruk Djodjic[2], Magdalena Bieroza[1]

[1] Department of Soil and Environment, Swedish University of Agricultural Sciences, Uppsala, Box 7014, 750 07, Sweden

[2] Department of Aquatic Sciences and Assessment, Swedish University of Agricultural Sciences, Box 7050, 750 07, Uppsala, Sweden

*Correspondence to*: Lukas Hallberg (lukas.hallberg@slu.se)

**Abstract.** Agricultural headwater streams are important pathways for diffuse sediment and nutrient losses, requiring mitigation strategies beyond in-field measures to intercept the transport of pollutants to downstream freshwater resources. As such, floodplains can be constructed along existing agricultural streams and ditches to improve fluvial stability and promote deposition of sediments and particulate phosphorus. In this study, we evaluated 10 remediated agricultural streams in Sweden for their capacity to reduce sediment and particulate phosphorus export and investigated the interplay between fluvial processes

and phosphorus dynamics. Remediated streams with different floodplain designs (either on one side or both sides of channel, with different width and elevation) were paired with upstream trapezoidal channels as controls. We used sedimentation plates to determine seasonal patterns in sediment deposition on channel beds and floodplains and monthly water quality monitoring. This was combined with continuous flow discharge measurements to examine suspended sediment and particulate phosphorus dynamics and reduction along reaches. Remediated streams with floodplains on both sides of the channel reduced particulate

phosphorus concentrations and loads (-54 µg L$^{-1}$, -0.21 kg ha$^{-1}$ yr$^{-1}$) along reaches, whereas increases occurred along streams with one-sided floodplains (27 µg L$^{-1}$, 0.09 kg ha$^{-1}$ yr$^{-1}$) and control streams (46.6 µg L$^{-1}$). Sediment deposition in remediated streams was five times higher on channel beds compared to floodplains and there was no evident lateral distribution of sediments from channel to floodplains. There was no effect from sediment deposition on particulate phosphorus reduction, suggesting that bank stabilization was the key determinant for phosphorus mitigation in remediated streams, which can be

realized with two-sided but not one-sided floodplains. Further, the overall narrow floodplain widths likely restricted reach-scale sediment deposition and its impact on P reductions. To fully understand remediated streams' potential for reductions in both nitrogen and different phosphorus species and avoid pollution swapping effects, there is a need to further investigate how floodplain design can be optimized to achieve a holistic solution towards improved stream water quality.

# 1 Introduction

Past and present inputs of bioavailable nutrients for agricultural production continue to negatively impact stream water quality and ecology across aquatic ecosystems globally (Sharpley et al., 2013; Smith, 2003). In particular, phosphorus (P) losses to surface waters play a key role in eutrophication since it often is the limiting element for algal and cyanobacterial growth in freshwater recipients (Correll, 1998). The transport of P from land to rivers and lakes is disproportionately influenced by headwater streams, representing entry points for sediments and nutrients to stream networks. Therefore, headwater streams' capacity for processing and removal of pollutants is critical for regulating downstream export (Wollheim et al., 2018). Headwaters in flat agricultural landscapes (< 2 % slopes) commonly comprise of artificially straightened and deepened trapezoidal channels (Fig. 1b) that effectively drain excess water from fields and enable crop growth. However, their short water residence time and geomorphic instability also enhance downstream export of suspended sediments (SS) and associated P. Therefore, an improved understanding of the capacity for sediment and nutrient reductions within agricultural stream networks is critical, and increasingly so with a changing a climate that accelerates pollution to freshwaters (Ho et al., 2019; Ockenden et al., 2017).

High export of sediments and P from trapezoidal channels, either historically converted from natural streams or artificially introduced, results from altered transport capacity and an imbalance between catchment's supply and stream's transporting power (Magner et al., 2012; Simon and Rinaldi, 2006). Internal erosion from trapezoidal channels can therefore contribute with higher sediment and P loads compared to distal sediment sources in catchments (Simon and Rinaldi, 2006). Closely interlinked with sediment conveyance in channels is the instream transport and storage of P, governed by fluvial processes (deposition/resuspension of sediment-bound P; Ballantine et al., 2009; Noe et al., 2019), redox processes (adsorption/desorption of soluble reactive P [SRP] to SS; Sandström et al., 2021) and geomorphic stability (streambank erosion; Fox et al., 2016). Channel bed sediments can store high amounts of P (Ballantine et al., 2009) and although deposition has been found to dominate in headwater reaches due to stream power limitation (Worrall et al., 2020), sediment-bound P becomes susceptible to remobilization during high flow events (Bowes et al., 2003). Streambank erosion due to bank failures in agricultural streams can also be a significant source of P export particularly in P-rich bank soils (Fox et al., 2016; Kronvang et al., 2013; Landemaine et al., 2015), but the efficacy of management strategies to counter this remains underexplored.

To reduce erosion and nutrient losses while maintaining drainage capacity, two-stage ditches with lateral floodplains excavated along existing channels have been tested as a stream remediation measure (Powell et al., 2007), with either two narrower or one single wider floodplain (Fig. 1). Here, floodplains refer to 2 to 6 m wide benches that are inundated upon higher flows (D'Ambrosio et al., 2015; Mahl et al., 2015; Powell et al., 2007). The purpose of two-stage designs is to establish a fluvial equilibrium by mimicking natural floodplain formation. Their design can support stable floodplains and banks, aid flood management and reduce the need for routine dredging and bank restoration (D'Ambrosio et al., 2015; Krider et al., 2017). During higher flows, stream power dissipates as water inundates vegetated floodplains, allowing for reductions in SS and P concentrations by sedimentation (Davis et al., 2015; Mahl et al., 2015), sediment adsorption of SRP (Trentman et al., 2020)

and nitrate removal through denitrification (Hallberg et al., 2022; Mahl et al., 2015; Speir et al., 2020). Reductions in SS and P concentrations and loads along remediated streams with a two-stage design are argued to be driven by lateral distribution of sediments from inset channel onto floodplains during higher flows (Fig. 1), based on monitoring studies (D'Ambrosio et al., 2015; Krider et al., 2017; Mahl et al., 2015) and flume experiments (Bai and Zeng, 2019). To date, studies on sediment delivery in remediated streams have solely relied on either geometric survey data or grab water sampling, with emphasis on investigating its potential for improving geomorphic stability. Therefore, there is a need to explicitly link net P deposition rates and fluvial dynamics in field conditions with observed water quality impacts.

We measured net sedimentation rates and concentrations of P fractions in channel bed and floodplains of 10 remediated streams (Fig. 2a) with two-stage design (Fig. 1) with the aim to investigate spatial and temporal distribution of sediments and the effect on P export and long-term storage. Changes in stream water P concentrations along remediated streams and upstream trapezoidal control streams were compared to determine their reduction efficacy (Fig. 2b). Our objectives were to determine 1) if remediated streams with two-stage design can reduce SS and P export, 2) how floodplain designs (elevation and one- vs. two-sided floodplains) affects reduction capacity and 3) what factors of transport, erosion and deposition that control SS and P reductions. Two types of floodplain designs were compared: one-sided and two-sided floodplains along with a gradient in floodplain elevations and thus inundation frequencies.

## 2 Materials and methods

### 2.1 Site description

Ten study sites were selected with established two-stage ditches (referred to as remediated streams) in Central East (C1–5) and South Sweden (S6–S10; Fig. 2a). The study sites are located in low slope gradient and tile drained agricultural catchments, dominated by winter and spring sown cereal crops and ley grass. A control-impact design was used to compare remediated reaches (0.3 to 1.7 km reach lengths) with upstream, mostly trapezoidal reaches, hereafter referred to as control streams (see Fig. S6–9 for comparison of profiles). An exception is site C4 where the erosion of stream bank has created a natural floodplain (Fig. S7). Control stream reaches were selected with equivalent lengths (where this was possible) and similar channel slopes and agricultural land use % in sub-catchments (Table S1). For detailed monitoring design see Fig. S1–5. Site S7 was not paired with a control reach since the remediated stream originates from a wetland. Additionally, sites S7 and S8 are nested in the same stream network (Fig 2a), with approximately 10 km stream length in between. Site C1 has two tributaries draining into the remediated stream reach.

In remediated streams, floodplains have been constructed either symmetrically on two sides (C1–3 and S8) or on one side (C4, S6–7 and S9–10) of the inset channel, site C5 being the exception with both designs along the reach. One-sided floodplains with double widths were constructed in streams where riparian obstructions (e.g., trees, boulders and high field elevation) restricted two-sided floodplain excavation. Floodplain elevations ranges between 0.25 and 0.96 m in relation to the lowest point of inset channel and the ratio of total flooded widths to channel top widths ranges between 1.3 and 2.9. Remediated

streams were constructed between 2013 and 2019, with the purpose to prevent flooding of adjacent fields and to improve fluvial stability and downstream water quality (Brink et al., 2012; Wiström and Lindberg 2016; Hedin and Kivivuori 2015).

The sites are variable in their catchment characteristics and hydrology (Table 1). Central East catchments (C1–5) are generally characterized by lower agricultural land use density and higher clay content in soils compared to South catchments (S6–10). Overall, the remediated streams are dominated by base flows, with median discharge ranging between 0.01 to 0.32 $m^3 s^{-1}$.

## 2.2 Sediment deposition rates: field sampling and analysis

To collect net sediment deposition rates, 0.16 $m^2$ square wooden fiber plates were installed in September 2020 and anchored with a 1 m metal rod through its center (Fig. S10). Plates with a smooth surface coating were chosen to allow for resuspension of settled sediments but also to enable rapid *in situ* sediment collection. Sedimentation plates were installed on channel beds and on floodplains along each remediated ditch at the three locations upstream (US), mid (MS) and downstream (DS; Fig 2b), except at site S6 and S8–9, where no plates could be placed in channels due to impenetrable gravel and pebble bed substrates. Over the course of 21 months, deposited sediments were collected from the plates three times, approximately biannually (Fig. 2c) and with a total sample count of 41 (channel) and 73 (floodplains). To collect intact sediment samples from submerged plates in channels, a cylindrical frame was fitted onto plates before retrieval to prevent losses of sediments to water column (Fig. S10). Before sampling, plates were lifted up from channel beds to dewater sediments and the sediments were then measured for depth to estimate the total volume. Sediments were only collected from the inner 0.04 $m^2$ square of the plates to avoid edge effects, and subsequently stored in airtight plastic bags. Sediment plates with high volumes (n = 32) were subsampled from the 0.04 $m^2$ square, with approximately 1 kg of fresh weight sediments collected after homogenization in field. The fresh weight per 0.04 $m^2$ of subsampled sediments was then estimated from the product of sediment volume on plate and bulk density of fresh sediments, measured per 100 ml. In case of missing sediment depth values in samples (n = 13), the depth was estimated based on a linear relationship between sediment fresh weight (kg $m^{-2}$) and sediment depth (mm) of known samples (n = 90; r = 0.99, p < 0.01). All samples were oven dried at 105 °C to determine dry matter and then analyzed for TP with ICP-OES (Avio 200, PerkinElmer, USA). The TP content in sediments were normalized to deposition rates per area over six months as well as one year.

## 2.3 Composite sediment sampling and sequential P fractionation in sediments

In addition to newly deposited sediments on plates, composite sediments that represent the integration of both short- and long-term sediment deposition, were sampled with a trowel in spring 2022 to determine P fractions. Sediments in channel bed and floodplains were sampled at US and DS for sequential P fractionation analysis (Fig. 2b, c). Sediments were sampled down to 5 cm depth with a trowel as 5 $cm^3$ cubes. At each channel location, five pseudo-replicates were sampled at 10 m distance along reaches and pooled into one sample. On floodplains, 10 pseudo-replicates were sampled at 5 m distance. Sediments were

placed in airtight plastic bags and stored in coolers during field transportation and at 4 °C back at laboratory before dry matter was determined with oven drying at 105 °C.

To determine the forms of P present in composite sediment samples, a sequential P fractionation method was used, developed from Psenner and Puckso (1988) and Hupfer et al. (1995, 2009). The analyzed P species (as operationally defined) were: water soluble P ($H_2O$-P), redox-sensitive P adsorbed to iron and manganese (Fe-P), $OH^-$ exchangeable P adsorbed mainly to aluminum (Al-P), P bound in organic compounds (Org-P) and calcium-bound P (Ca-P). Here, $H_2O$-P and Fe-P are also referred to as labile P and Al-P, Org-P and Ca-P as recalcitrant P. The residual non-reactive P after aforementioned fractionation (refractory P) was not determined; cumulative concentrations of analyzed P fractions are here referred to as TP. Fresh sediment samples were sequentially extracted with Milli-Q water ($H_2O$-P), buffered dithionate solution (Fe-P), NaOH (Al-P), NaOH after 30 min persulfate digestion at 120 °C, subtracted with NaOH (Org-P), and HCl (Ca-P). All extractions were analyzed for unfiltered SRP using the molybdenum blue method (Murphy and Riley, 1962). Concentrations of P fractions in sediments were calculated as g P $kg^{-1}$ DM for comparison with TP concentrations in newly deposited sediments.

## 2.4 Hydrological monitoring and stream power calculation

Discharge was estimated by establishing stage-discharge rating curves from discharge measured manually at four to eight occasions and continuous water stage data from pressure sensors, as described in Hallberg et al. (2022). To minimize overestimation of high flows outside the range of flow measurements, out-of-range flows were estimated for all sites except C2 and C4, using the product of 1) polynomial regressions of stage-cross-section area and 2) logarithmic regressions of stage-mean velocity (Herschy, 2014). To determine the effect of floodplain inundation on sediment deposition, geometrical cross-section surveys were conducted and floodplain inundation events were subsequently estimated using stage data (Hallberg et al., 2022).

The power generated by a stream is a function of the amount of water that flows down a slope in a confined channel. To compare the stream power and its influence on bed sediments between streams of different sizes (Bagnold, 1973; Reinfelds et al., 2004), the total stream power can be normalized by channel width to unit stream power (ω) at a given cross-section using:

$$\omega = \frac{\rho g Q S}{w} \tag{1}$$

where $\rho$ is the density of the water (kg $m^{-3}$), $g$ is the acceleration due to gravity ($m^2$ $s^{-1}$), $Q$ is the flow discharge ($m^3$ $s^{-1}$), $S$ is dimensionless channel bed slope and $w$ is the channel width (m), here measured as the top width of the inset channel. With unit stream power, the force exerted per surface area and thus potential sediment transport can be derived, irrespective of system size. Unit stream power was calculated at point of floodplain overflow, using derived flow discharge and channel widths from geometrical survey, representing the maximum force of water when restricted to inset channel.

## 2.5 Water quality sampling and vegetation survey

Water grab samples were collected monthly between April 2020 to July 2022 to determine water chemistry at the four locations upstream control (TD), US, MS and DS (Fig. 2b). At site S9, sampling after June 2021 was excluded from analysis since the control stream was then converted into a remediated stream. Water samples were analyzed for TP (SS-EN ISO 6878:2005), with and without 0.45 µm filtration, and suspended solids (SS; SS-EN 872:2005). Particulate P was calculated as the difference between unfiltered and filtered TP. Existing TP, PP and SS water chemistry from a location 1 km downstream of site C3 (Kyllmar et al., 2014) were used for subsequent validation of flow-weighted mean concentration (FW) load estimation method. Two types of data were used: 1) Annual loads estimated from fortnightly flow-proportional composite sampling and 2) fortnightly discrete concentration samples. In addition, vegetation cover was visually estimated on floodplains using three 1 m$^2$ (1 x 1 m) plots at each of the three locations US, MS and DS. Average percent vegetation cover was calculated from two surveys in May and June 2021.

## 2.6 Statistical data analysis

All statistical analyses were performed in R version 4.2.1 (RStudio Team, 2022). The packages hydrostats (Bond, 2022) and ContDataQC (Leppo, 2023) were used to analyze the hydrological regimes. Base flow index (Gustard et al., 1992) were calculated using *baseflows* and flashiness index (Baker et al., 2004) using *RBIcalc*. Two outliers in TP and PP concentrations and one outlier in SS concentrations were removed from further analysis; these data points had exceptionally high concentrations (studentized residuals > 10, i.e., regression model residual divided by its adjusted standard error) and were measured in stagnant water during summer months, with minimal influence on total loads.

Annual loads of TP, PP and SS per ha were calculated at US and DS locations using FW method (Elwan et al., 2018). The accuracy of annual load estimation was tested downstream of site C3. Here, loads calculated from existing fortnightly flow-proportional composite sampling represent 'accurate' loads (Dialameh & Ghane, 2022). At the same location, FW loads were calculated for the study period using water samples from two to three different days within each month to compare with true loads (Fig. S11). At this location, annual FW loads were underestimated across all parameters (TP: -28 %, PP: -29 % and SS: -52 %). Daily loads of TP based on monthly water chemistry were also calculated by multiplying with daily flow. Changes in TP, PP and SS concentrations were calculated as the difference between US and TD (control streams) and the difference between DS and US (remediated streams), hence negative values represent reductions in concentrations along reaches. In the case of non-normal sample distributions of changes in TP, PP and SS concentrations (i.e., slightly right-skewed and with tails of outliers to both left and right), we used the approach of non-parametric permutation of t-tests (R = 10,000), using MKinfer package (Kohl, 2022). Permutation tests do not assume an underlying distribution and solely rely on the assumption of exchangeability, such as the absence of temporal autocorrelation (Good, 2000). To test for autocorrelation, concentration data was regressed against date and, if significant ($p < 0.05$), it was excluded from subsequent permuted t-test. Differences in changes in TP, PP and SS concentrations between remediated and control streams as well as differences in P deposition rates

across inundation classes and floodplain sides were tested with permuted unpaired two-tailed t-tests (*perm.t.test*). When testing differences in concentration changes for each individual site, paired two-tailed t-tests (*t.test,* stats package; 'base' R) were used for normally distributed data and permuted paired two-tailed t-tests (*perm.t.test*) were used for non-normal data. Seasonal differences in P deposition rates were tested with one-way ANOVA (*anova*, stats).

The effects of P deposition and stream water P on labile P in composite sediments were quantified with Pearson correlation coefficients, using *cor.test* in stats package. No permutation of Pearson correlations was necessary due to met assumptions of homoscedasticity and normal distribution of residuals. To visualize the controls for P sedimentation, sample distribution of predictor variables (catchment and reach properties) was analyzed using scaled principal component analysis (PCA; *rda*) on two separate matrices for channels and floodplains, using the Vegan package (Oksanen et al., 2022). Vectors of P deposition rates were fitted to the PCAs using *envfit* with 10,000 permutations.

## 3 Results

### 3.1 Suspended sediment and P reduction in stream water

Absolute concentrations of TP and PP in stream water were highly variable across both seasons and sites, peaking in summer months during low flows and ranging from below detection limit to 3940 µg TP $L^{-1}$ and 3557 µg PP $L^{-1}$. Overall, the proportion of PP out of TP ranged from 28 % to 86 % but PP was the dominant form across seven sites (Table S2). Concentrations of SS ranged between < 1 to 1200 mg SS $L^{-1}$ and were correlated to PP concentrations at each site (Table S3), but to a lesser degree in site C1 and S9 at high concentrations. Changes in TP concentrations along upstream control reaches correlated weakly with base flow index ($r = 0.27$, $p < 0.01$) and flashiness index ($r = -0.25$, $p < 0.01$), but there was no general correlation along remediated reaches and thus little influence of different hydrological regimes on P concentrations dynamics.

When grouping remediated streams by floodplain design, two-sided floodplains showed higher reductions in absolute TP and PP concentrations along reaches, compared to control streams (Fig. 3). Concentration changes along remediated streams with one-sided floodplains were not different compared to control streams. Correspondingly, there was a similar pattern to changes in TP and PP loads along remediated streams: two-sided floodplains retained on average $0.17 \pm 0.26$ kg TP $ha^{-1}$ $yr^{-1}$ and $0.21 \pm 0.21$ kg PP $ha^{-1}$ $yr^{-1}$ while one-sided floodplains lost $0.07 \pm 0.10$ kg TP $ha^{-1}$ $yr^{-1}$ and $0.09 \pm 0.10$ kg PP $ha^{-1}$ $yr^{-1}$. Concentration reductions in TP and PP among sites with two-sided floodplains were mostly explained by site C2 but also S8 (Fig. 3). Changes in SS concentrations were not reduced along two-sided floodplains and site C3 significantly increased SS concentrations along remediated reach.

Based on monthly water chemistry data, TP loads (g $day^{-1}$ $ha^{-1}$) were reduced during inundation events in site S8, compared to base flows (permuted t-test, $p = 0.02$) . This effect was not observed at any other site. Higher incoming loads in two-sided floodplains led to greater load reduction, while higher loads in one-sided floodplains led to greater load increases along the remediated reaches. (Fig. 4).

## 3.2 Deposition of sediments and P in channels and on floodplains

Sediment deposition depth was five times higher on channel beds ($86 \pm 72$ mm yr$^{-1}$) compared to floodplains ($16 \pm 23$ mm yr$^{-1}$) at sites C1–5, S7 and S10 (Fig. 5a), showing that sedimentation on floodplains, despite inundation frequencies up to 200 days yr$^{-1}$, contributed little to the reaches' total sediment budgets. Deposition rates of P were correlated to sediment depth (r = 0.96, p < 0.01) and averaged $29.9 \pm 29.3$ g TP m$^{-2}$ yr$^{-1}$ on channel beds and $7.0 \pm 10.4$ g TP m$^{-2}$ yr$^{-1}$ on floodplains. Combined channel bed and floodplain deposition rates did not correlate with stream water P concentration changes, neither between remediated and control streams and irrespective of number of floodplain sides (Fig. S12). Ratios of deposited P:sediment mass did not differ between channel and floodplains (unpaired t-test, p = 0.83). There was a seasonal pattern of higher P deposition rates on channel beds during the more hydrologically active winter and spring seasons, compared to summer and autumn (ANOVA, $F_{2, 38}$ = 2.86, p = 0.07).

On channel beds, P deposition rates increased with lower unit stream power (r = -0.43, p = 0.03; Fig. S13), representing the maximum stream power per m$^{-2}$ bed surface area when flow was confined to the inset channel. Deposited P did not correlate with the average of stream water TP loads for each 6 month sampling period (r = -0.16, p = 0.42; Fig. S13) and neither with the ratio of unit stream power:TP loads (r = 0.28, p = 0.32), indicating that unit stream power alone during lower flows was the primary driver for P sedimentation on channel beds.

On floodplains, P deposition rates fitted to a PCA of environmental predictors were primarily controlled by inundation frequency (Fig. S13b), in accordance with linear regression (Fig. 5c). Vegetation cover was also dependent on inundation frequency, but when excluding data points < 70 inundation days yr$^{-1}$, there was an effect of higher P deposition rates with vegetation cover, independent of inundation (adj. $R^2$ = 0.05, Vegetation p = 0.05, Inundation p = 0.87). In contrast to channel beds, unit stream power did not influence P deposition rates on floodplains. Grouping of sites by mean inundation frequency revealed that P deposition rates on floodplains were 10.7 times higher during > 90 inundation days yr$^{-1}$ compared to < 20 inundation days yr$^{-1}$ (Fig. 5b). However, the process of lateral distribution from channel to floodplains could not be confirmed as P deposition rates on channel beds were not influenced by inundation.

## 3.3 Influence of sediment deposition on P stocks

There was an increase in TP content of composite sediments with construction age of remediated streams, both in channel beds (r = 0.48, p = 0.04) and floodplains (r = 0.54, p = 0.02), which was best explained by increases in Al-P fraction. Labile P fractions (H$_2$O-P and Fe-P) accounted for on average 19 % of TP, hence the recalcitrant P fractions (Al-P, Org-P and Ca-P) were the dominant forms of P in sediments (Fig. S14). There was a trend in higher labile P content (paired t-test, p = 0.06) in channel sediments compared to floodplain sediments but no differences for TP or recalcitrant P content (paired t-test, TP: p = 0.12, Recalcitrant P: p = 0.23).

Total P in newly deposited sediments on plates was 1.6 times higher than in composite sediments, i.e., sediments down to 5 cm depth outside of plates. Newly deposited P further correlated with higher labile P content in composite sediments of

channels but not floodplains (Fig. 6a). However, deposited P had no effect on recalcitrant P content, meaning that deposited P mainly contributed to the bioavailable P pool in channel sediments (Fig. 6b). Stream water concentrations of TP and PP predicted higher labile P content in channel and floodplain sediments equally well (Fig. 6c, d), indicating that P in water column supply bioavailable P on floodplains rather than sediment deposition.

## 4 Discussion

### 255 4.1 Floodplain design controls P reduction in remediated streams

The observed reductions in TP and PP concentrations and loads along remediated streams with two-sided floodplains show that this measure has the potential for improving downstream water quality when compared to control streams. The effect was exclusive to two-sided floodplain designs; one-sided floodplains did not differ from control streams, which implicates that a floodplain on only one side is less suitable for P mitigation. Differences in P loads between the two floodplain designs were

even more pronounced in sites with higher catchment P supply, as two-sided floodplains responded as active P sinks whereas one-sided floodplains behaved as passive conduits. However, P load data were only available for two sites with two-sided floodplains which limits the certainty of this floodplain design being the main control responsible for P load reductions, warranting further study of floodplain designs. Further, flow-weighted average concentration load estimation is generally conservative (Elwan et al., 2018), which also was shown in proximity of site C3. Therefore, the annual TP and PP loads may

in fact be underestimated with 30 %. Remediated streams with lower catchment P loads, due to less agricultural land use, did not reduce P loads irrespective of floodplain design. Thus, to realize their P reduction potential, the results indicate that remediated streams should be targeted in areas with P loading exceeding 0.25 kg P ha$^{-1}$ yr$^{-1}$.

Our results suggest that PP reductions can be substantial in these systems and lead to decreased TP loads, adding to previous body of research from the US (Davis et al., 2015; Kindervater and Steinman, 2019; Mahl et al., 2015). This is particularly

important in clay-dominated catchments, associated with high PP delivery from soil erosion to streams (Sandström et al., 2020) as in site C2. The highest reductions in both TP (30 %) and PP (34 %) concentrations occurred along the remediated stream in site C2, characterized by silty clay loam soil texture. These reductions were in addition contributed by a perched culvert placed at the outlet, resulting in pre-dominantly stagnant water conditions at the lower end of the reach. In combination with little flow accumulation, this resulted in highly favourable conditions for P trapping, without the risk of bank overflow. Although

the majority of the studied catchments were dominated by PP, site S8 which (in contrast to PP-dominated C2) was characterized by equal proportions of PP and SRP and loam soil texture, also showed a tendency of reductions in TP, PP and SS concentrations. This implies that remediated streams can be effective across diverse catchments with contrasting soil textures and P forms, if the criteria of high P loading is met. In addition, site S8 showed that remediated streams with frequent inundation and without additional ponding, as in C2, can improve downstream water quality. Site S8 has also been subject to

re-meandering which could account for increased water residence times and settling of particulates, in addition to its floodplains. However, among two-sided floodplain sites, C1 and C3 did not reduce P and SS, implying that solely

implementing two-sided floodplains is insufficient and that further requirements such as frequent inundation, floodplain stability and reach lengths must be met by proper design of two-sided sites. For instance, site C1 is characterized by a short and straight reach with low flooding to channel ratio and unstable floodplains that prevent P and SS reduction.

We assumed that lateral inputs (tile drains and groundwater inflow) along both remediated and control reaches were comparable and thus not influencing the comparison between the two reaches. All paired reaches received tile drain inputs from adjacent fields with identical crop cultivation and flat topography with predominant subsurface flow pathways. There were no large deviations in loads between up- and downstream and we do therefore not have any reasons for suspecting significant lateral hotspots.

Our study confirmed that it was during inundation events that reductions in P loads occurred along streams with two-sided floodplains, consistent with previous research that have emphasized the importance of periodic floodplain inundation as master variable for reducing SS and P losses via flow dissipation (Bai and Zeng, 2019; D'Ambrosio et al., 2015; Krider et al., 2017). The lack of P reductions during inundation along streams with one-sided floodplains was likely connected to higher erosion from banks next to inset channels. These banks had higher slopes and resembled trapezoidal banks, compared to banks adjacent

to floodplains (data not shown; Fig. 1). Higher bank slopes in loam and clay soils are known to reduce the cohesive strength and increase the vulnerability towards erosion of bank material to the water column (Thorne and Tovey, 1981). In one-sided designs, banks next to inset channels were also exposed to higher stream velocities; the greater water depth in channels compared to floodplains led to higher water velocities that exerted higher shear stress on banks adjacent to the channel. Accordingly, we suggest that these processes that drive bank erosion together can explain the lack of P reductions with one-

sided floodplains, consistent with previous observations of increased scour and deposition with this floodplain design (D'Ambrosio et al., 2015).

## 4.2 Influence of sediment deposition on P dynamics and channel morphology

Contrary to our initial assumption, higher P deposition rates on floodplains did not reduce P concentrations in stream water and could therefore not be confirmed as the sole process responsible for SS and P reduction in remediated streams. Instead,

there was an average increase in TP concentrations of 1.6 µg TP $L^{-1}$ per deposited g TP $m^{-2}$ $yr^{-1}$, suggesting that floodplain deposition co-occurred with other processes contributing to P losses e.g., resuspension. Deposition rates of sediments and P on floodplains were controlled by inundation frequency and were substantially higher in this study compared to reported rates from five US sites (0.5–13 mm sediment $yr^{-1}$; D'Ambrosio et al., 2015) and three Swedish sites (0.04–1.2 g TP $m^{-2}$ $yr^{-1}$; Lacoursière and Vought, 2020).

The relative importance of vegetation cover for trapping sediment-bound P on floodplains could only be assessed in locations with > 70 days of inundation $yr^{-1}$, without collinearity between the two variables. Provision of water and nutrients from inundation events increases biomass growth on floodplains. It can therefore be assumed that sediment deposition is promoted directly by inundation, that provide sediments to be settled, and indirectly by its effect on vegetation that also traps

sediments. However, the smooth surfaces of sediment plates that were used to measure deposition likely underestimated sediment trapping on densely vegetated floodplains.

Despite the relatively high sediment yields on floodplains, particularly at sites C4, S6 and S10, its lack of effect on P concentrations observed in our study contradicts previous findings, reporting floodplains as consistent P traps (D'Ambrosio et al., 2015; Davis et al., 2015; Mahl et al., 2015). An explanation for this could be that in our study sites, flooding to channel width ratios were significantly smaller than the recommended ratio of 3 to 5 (Powell et al., 2007) and therefore not contributing to sufficient reach-scale P deposition needed to impact P in the water column. The unexpected increase in P concentrations with floodplain deposition could be due to coinciding activation of erosion processes in either channel, banks, floodplains that override reductions from sedimentation. Total P losses during increasing floodplain deposition, were primarily explained by mobilization or increased transport of PP, accounting for 87 % of TP losses, and not SRP. Overall, surface erosion in catchment soils did not drive stream sediment losses. Modelled maximum erosion rates of SS from surface runoff (Djodjic and Markensten, 2019) were generally low for all sites (10−50 kg ha$^{-1}$ yr$^{-1}$), except for site S7 (100−250 kg ha$^{-1}$ yr$^{-1}$), which likely contributed to SS and PP loading in that particular site. It is further possible that rapid resuspension of newly deposited sediments during higher flows lead to net losses of stream water P. As monitoring with sediment plates only accounted for net deposition, we could not quantify losses of sediments co-occurring with deposition.

Based on sediment deposition rates on plates, there was no indication of lateral sediment distribution from channel beds to floodplains in the seven sites (C1–5, S7, S10) where both channel beds and floodplains were evaluated. Instead, channel beds represented the dominant pool of sediment storage in remediated streams, independent of floodplain deposition rates. The ongoing aggradation on the channel beds indicate that these systems are still in fluvial disequilibrium, which is likely an indirect effect of floodplain inundation that also lower water velocities in channels. However, we could not show if this also occurred in control streams since no channel beds were monitored in these reaches. The lack of correlation between channel deposition and reductions in P concentrations in stream water could be explained by the bi-annual sediment sampling interval, being too coarse to capture periodic P reduction from sedimentation, which has been reported to occur during lower flow conditions (Bowes et al., 2003). In sites S6 and S8-9, where channel slopes ranged between 0.2 to 0.6 %, channel bed material was dominated by gravel and pebble substrates. Their impenetrable channel beds prevented plate installation and measurement of sedimentation rates but based on visual observations, no substantial deposition of fine sediments occurred over the study period. Hence these channels appeared to have sufficient stream power in relation to SS loads to transport SS, driven by their higher longitudinal channel slopes.

## 4.3 Variability in sediment P stock

The dominance of P deposition on channel beds compared to floodplains were also reflected in the higher labile P fractions ($H_2O$-P and Fe-P) in composite channel bed sediments, which increased with P deposition rates. Although labile P fractions only accounted for 20 % of TP in composite sediments, Fe-P (the largest fraction of labile P) is sensitive to reducing conditions and subsequent re-release as SRP into the water column (Jarvie et al., 2005). The contribution of labile P from sediment

deposition can therefore have negative implications for P export, supplying the system with easily available P rather than providing a persistent P trap. Labile P in composite sediments also correlated with stream water P concentrations, but here the causation was less clear due to the reciprocity between sediment-bound P and stream water P; higher P concentrations in stream water can increase sediment P (deposition and sorption) while high sediment P also can lead to higher stream water P (resuspension and desorption).

The range of TP content in channel bed sediments was consistent with other agricultural streams in Sweden (0.07–1.57 g P kg$^{-1}$, Lannergård et al., 2020; 0.35–1.85 g P kg$^{-1}$, Sandström et al., 2021), indicating that remediated streams do not promote P enrichment in channel beds compared to streams without floodplains. However, the proportions of labile P fractions were overall lower in our study, either explained by less deposition of these fractions or a more rapid turnover of labile P. In a comparable system, sediments were also dominated by recalcitrant fractions that were shown to be stable over seasons (Kindervater and Steinman, 2019). The increase in Al-P with construction age was partly explained by higher clay content in catchment soils, which often are rich in Al silicates and hydroxides that increase the sorption of P to these surfaces (Gérard, 2016). However, the effect on TP with construction age was independent from clay content, suggesting a true accumulation of P in sediments over time in both channel and floodplain sediments. This post-construction maturation process indicate a system reset to low P content in sediments following construction, as previously shown for denitrification and microbial respiration rates in floodplain sediments (Speir et al., 2020).

The higher vegetation coverage on floodplains compared to channel beds was not reflected in higher organic P fraction in sediments. Although soil organic carbon have been found to accumulate more over time in floodplain sediments, compared to channel sediments (Hallberg et al., 2022; Speir et al., 2020), the lack of increase in organic P in floodplain sediments with construction age suggests that the build-up of organic P forms, despite presence of vegetation, is slow and require more time than 10 years to occur. In this study, P in vegetation biomass was not considered and although it could contribute as a seasonally active P sink, Kindervater and Steinman (2019) reported that channel and floodplain sediments on average held 18 times more P mass than biomass on reach-scale.

## 4.4 Floodplain management towards P reduction and fluvial equilibrium

Currently in Sweden, the recommended dimensions of remediated streams with two-stage design (1 km length, 3–5 flooding to channel width ratio) are solely based on hydrology and do not take into account impact on sediment or nutrient reduction (Lindmark et al., 2013; Powell et al., 2007). Moreover, subsidies from Swedish authorities compensate for the stream length when constructing remediated streams and there is no mechanism for incentivizing the implementation of wider floodplains, resulting in narrower than recommended widths in remediated streams. We suggest that wider floodplains and shorter lengths can have larger effects on P reductions by sedimentation, improving cost-efficiency while maintaining the same total area. It is therefore necessary to also evaluate this within a modelling framework that can isolate a range of dimensions from other confounding, site-specific factors.

A one-sided floodplain design is sometimes preferred among landowners when constructing floodplains due to lower cost,
easier implementation and future maintenance accessibility, especially in locations where riparian obstructions (e.g., trees, boulders and high bank elevation) inhibit two-sided design. However, our results show clearly that this design is insufficient to reduce P concentrations and loads compared to control streams and therefore should be avoided. Instead, priority should be given to establish two-sided floodplains in critical source areas with high P loading, where riparian conditions as well as channel slope allow.

Reductions in P loads explicitly occurred during inundation events, but there was no correlation between P reductions, either in concentrations or loads, and annual inundation frequencies. Since higher inundation frequency (provided by lower floodplain elevations) increased P deposition, it is possible that lower and wider floodplains can increase reach-scale P trapping and thereby reduce P exports. However, lower floodplains can be exposed to higher shear stress and risk of toe erosion of floodplains that can explain the lack of observed P reductions with deposition. When choosing the optimal floodplain elevation,
multiple pollutants (e.g., nitrogen; Hallberg et al, 2022) should be considered to avoid potential pollution swapping (Bieroza et al., 2019).

Combining floodplains with partial damming as in site C2, can lead to higher reductions in SS and P concentrations. Littlejohn et al. 2014 reported 45 % TP load reductions with two low-grade weirs in streams with constructed floodplains, comparable to that of site C2. Thus, raising culverts or installing low-grade weirs can complement remediated streams. The
395 increased bankfull volume with constructed floodplains reduces the risk of flooding when restricting outflows, however, hydraulic calculations should be made to ensure that large flood volumes are confined within the bankfull volume when installing impoundments. Although the combination of re-meandering and floodplains resulted in lower SS and P export in site S8, stable meanders can be difficult to establish without additional protective measures to avoid erosion and migration (Wohl et al., 2015). Results on P mitigation with re-meandered channels are scarce but it has been shown to improve stream
habitat quality (Lorenz et al., 2009). However, meanders rely on a certain freedom of channel migration that can infringe on productive agricultural land and also lead to higher sediment (and thus P) exports and are therefore less suitable in narrow riparian buffers with limited room for channel evolution.Sediment accumulation on channel beds is widely perceived as negative by landowners, requiring routine channel dredging to avoid drainage impairment and to reduce the risk of flooding. However, in the absence of maintenance activities, natural formation of floodplains in agricultural streams have been observed
to develop from sediment deposition in channels, facilitated by over-widened channels that limit stream power and bed load transport (Jayakaran and Ward, 2007; Landwehr and Rhoads, 2003). As such, the predominant accumulation of channel bed sediments in our studied sites could indicate that there is an ongoing adjustment towards floodplain formation, in addition to already existing floodplains. The room for allowing channel sedimentation before it puts conveyance capacity at risk is also higher in remediated streams and could motivate abstention from dredging to allow stabilization of fluvial equilibrium and
adjustments of floodplains to the streams' fluvial conditions. Enhancing this potential by providing more space and designing wider floodplains is therefore important to effectively reduce P exports, rather than resorting to frequent channel dredging that systematically resets fluvial imbalance. Dense vegetation on floodplains of remediated streams also offer opportunities for

removing P accumulated in biomass, as dry floodplains are more accessible for mowing and collection compared to submerged macrophytes in channel. However, woody vegetation on banks and floodplains can physically obstruct operations to remove biomass.

As future climate will bring higher frequencies of hydrological extremes and consequently higher P exports (Mehdi et al., 2015; Ockenden et al., 2017), remediation of agricultural streams can offer improved flood control together with enhanced water purification.

## 5 Conclusion

Human pressures on freshwater resources, exacerbated by climate change, require urgent adjustment in agricultural streams to improve their capacity to transport and process large water and pollution fluxes. To date, nutrient mitigation strategies focus largely on in-field and point measures (e.g., soil management and wetlands), ignoring the enormous potential of headwater agricultural streams and their corridors as interfaces for reducing multiple pollutants (SS, N and P) while also providing flood control. Evaluating the capacity for stream remediation in agricultural landscapes to reduce SS and P export, as carried out in this study, is therefore a critical step for improving our understanding of how agricultural streams can be modified to reduce downstream water quality impacts.

In this study, we demonstrated that remediated streams reduce stream water P concentrations and loads while simultaneously increasing water conveyance capacity. Reductions in P were controlled by floodplain design, as floodplains constructed laterally on both sides reduced P exports during inundation events, compared to one-sided floodplains and control streams (trapezoidal channels). Higher catchment supply of P, driven by agricultural land use, also increased P reduction capacity in two-sided remediated streams. By linking fluvial dynamics with P storage and water quality impacts, our results showed that with narrow floodplains, PP reductions were controlled rather by protection against erosion than promotion of deposition. In this sense, remediated streams operate on a different principle than flow-through wetlands, with the focus on reducing erosion and near-stream P inputs rather than actively trapping upstream-derived sediments. However, by extending the width of floodplains, the importance of P deposition for PP reductions is expected to increase. The fact that heavily impacted catchments in this study delivered high loads of N and P, as is typical in agricultural streams, underscore the need for adopting management strategies that can maximize reductions of multiple pollutants and minimize the risk for pollution swapping. Although the optimal design for PP reduction also has the potential for promoting N removal, there is a further need for understanding the effect on SRP dynamics when promoting high inundation frequencies and deposition of sediments with labile P.

**Data availability**

The water and sediment chemistry and hydrology measurements from the studied sites are provided as Supplement.

## Supplement

The supplement related to this article is available online at: *DOI to be retrieved from HESS*

## Author contribution

Lukas Hallberg: Conceptualization, Methodology, Data curation, Formal analysis, Visualization, Writing – original draft, Writing - review and editing. Faruk Djodjic: Methodology, Formal analysis, Writing - review and editing. Magdalena Bieroza: Conceptualization, Methodology, Formal analysis, Writing - review and editing, funding acquisition.

## Competing interests

The authors declare that they have no conflict of interest.

## Acknowledgements

This project was funded by Svenska Forskningsrådet Formas (2018-00890), Havs- och vattenmyndigheten (3280-2019) and Oscar och Lili Lamms Minne (DO2019-0021) awarded to M. Bieroza. The authors would like to thank private landowners and stakeholders in the study catchments for their help with collecting water and sediments samples and providing access to field sites, especially Christoffer Bonthron from Tullstorpsåprojektet, Anuschka Heeb from Lovang AB and Dennis Wiström from Västerviks kommun. We would like also to thank Sheryl Illao Åström for conducting the geometrical surveys and compiling GPS data and Emma Ryding for collecting sediments and performing sequential P fractionation.

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

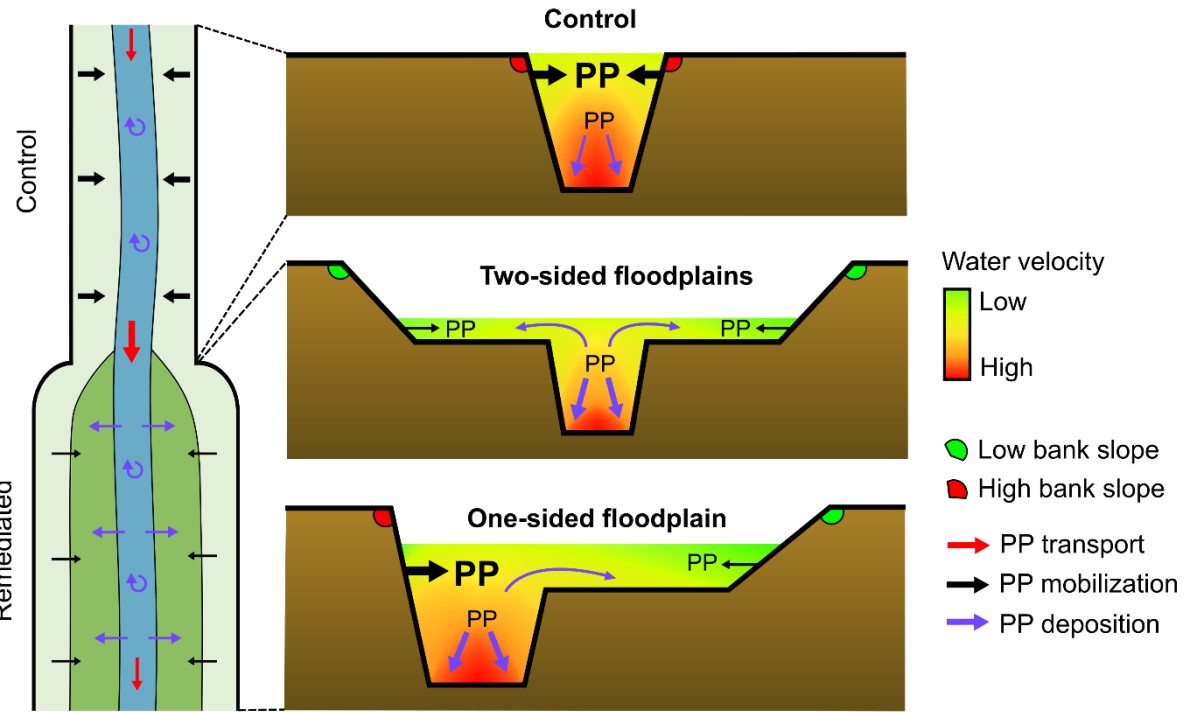

**Fig. 1. Conceptual model of fluvial processes that determine particulate phosphorus (PP) export in remediated vs. control streams. Deposition of PP occur on channel beds predominantly during lower flows when transport capacity are lower than sediment delivery. During inundation, PP deposition dominates on floodplains, due to lower water velocities. In two-sided floodplains, streambanks are stabilized during higher flows due to their lower slopes and lower shear stress from water forces compared to the inset channel. In one-sided designs, one streambank is steeper and exposed to higher water velocities in the channel, resulting in higher mobilization**
**and transport of PP.**

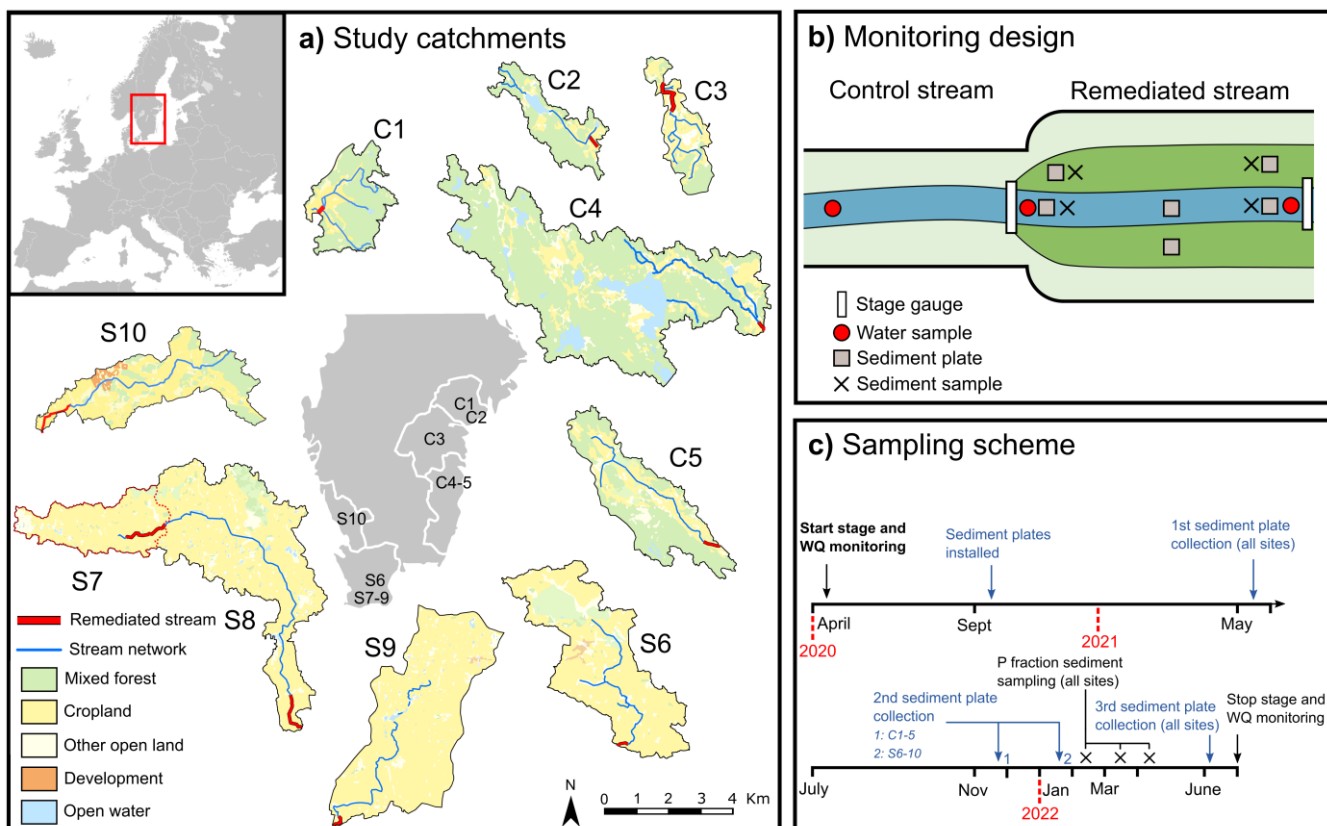

**Fig. 2.** Location of the 10 catchments with remediated streams in Central East (C1–C5) and South Sweden (S6–S10). b) Control-impact monitoring design. Sediment samples and water grab samples were collected at start, middle and end of remediated reaches and water grab samples were also collected at start of control reaches. Water stage sensors were placed at start and end of remediated reaches. c) Sampling scheme over time of sediments deposited on plates and composite sediments for sequential P fractionation. Land use maps: ©Lantmäteriet.




**Table 1. Characteristics of catchment, hydrology, ditch geometry of remediated streams. Sites are ordered from North to South. Flow discharge and floodplain inundation were recorded in April 2020 to June 2022. Discharge in site C3 was recorded between 2018 and 2022, interrupted between December to March each year. Ditch and floodplains geometry were measured in March and April 2021 with a GPS levelling device (E600 GNSS Receiver, E-survey, China).**

| | | Catchment properties | | | | Hydrology | | | | | | Year of construction | Ditch geometry | | | | | |
|---|---|---|---|---|---|---|---|---|---|---|---|---|---|---|---|---|---|---|
| Site | County | Area (km$^2$) | Precipitation (mm)* | Soil texture | Agricultural land use (%) | Q50 (m$^3$ s$^{-1}$) | Baseflow Index (BFI) | Flashiness Index (RBI) | Unit stream power** (W/m$^2$) | Channel max flow (m$^3$ s$^{-1}$) | Floodplain inundation (day yr$^{-1}$) | | Two-stage length (m) | Floodplain Height (m) | Floodplain width (m) | Flooding-Channel ratio | Channel slope (%) | Floodplain design |
| *Central East* | | | | | | | | | | | | | | | | | | |
| C1 | Södermanland | 9.75 | 594 | Silty clay | 15 | 0.02 | 0.80 | 0.43 | 0.05 | 0.04 | 107 | 2012 | 340 | 0.55 | 3.1 | 1.6 | 0.04 | 2-sided |
| C2 | Södermanland | 7.91 | 594 | Silty clay loam | 27 | 0.03 | 0.93 | 0.13 | 0.16 | 0.20 | 17 | 2012 | 730 | 0.96 | 3.5 | 2.2 | 0.04 | 2-sided |
| C3 | Östergötland | 6.63 | 551 | Clay loam | 70 | 0.01 | 0.82 | 0.30 | 0.38 | 0.03 | 17 | 2014 | 1500 | 0.53 | 3.2 | 2.5 | 0.30 | 2-sided |
| C4 | Kalmar | 45.5 | 627 | Clay loam | 16 | 0.32 | 0.91 | 0.12 | 1.29 | 0.56 | 187 | 2019 | 320 | 0.65 | 1.7 | 1.3 | 0.10 | 1-sided |
| C5 | Kalmar | 16.3 | 627 | Clay loam | 38 | 0.06 | 0.83 | 0.30 | 1.09 | 0.21 | 53 | 2012 | 780 | 0.65 | 1.4 | 1.5 | 0.14 | 1-sided/2-sided |
| *South* | | | | | | | | | | | | | | | | | | |
| S6 | Skåne | 22.7 | 707 | Loam | 77 | 0.09 | 0.83 | 0.35 | 1.43 | 0.13 | 100 | 2016 | 400 | 0.48 | 5.7 | 2.9 | 0.17 | 1-sided |
| S7 | Skåne | 10.8 | 569 | Loam | 81 | 0.04 | 0.81 | 0.30 | 0.23 | 0.07 | 159 | 2013 | 1960 | 0.55/0.25 | 4.0 | 2.8 | 0.09 | 1-sided |
| S8 | Skåne | 42.4 | 569 | Loam | 81 | 0.22 | 0.86 | 0.23 | 10.90 | 0.68 | 93 | 2013 | 1770 | 0.48 | 6.4 | 2.4 | 0.47 | 2-sided |
| S9 | Skåne | 31 | 569 | Loam | 86 | 0.11 | 0.90 | 0.09 | 15.15 | 2.18 | 6 | 2019 | 630 | 0.71 | 4.6 | 2.3 | 0.44 | 1-sided |
| S10 | Halland | 16.4 | 823 | Sandy loam | 58 | 0.20 | 0.89 | 0.20 | 1.04 | 0.34 | 123 | 2014 | 1760 | 0.34 | 2.8 | 2.2 | 0.09 | 1-sided |

*Mean annual rainfall 2016-2021 (https://www.smhi.se/data/meteorologi/nederbord).

**Unit stream power at the point of inset channel overflow

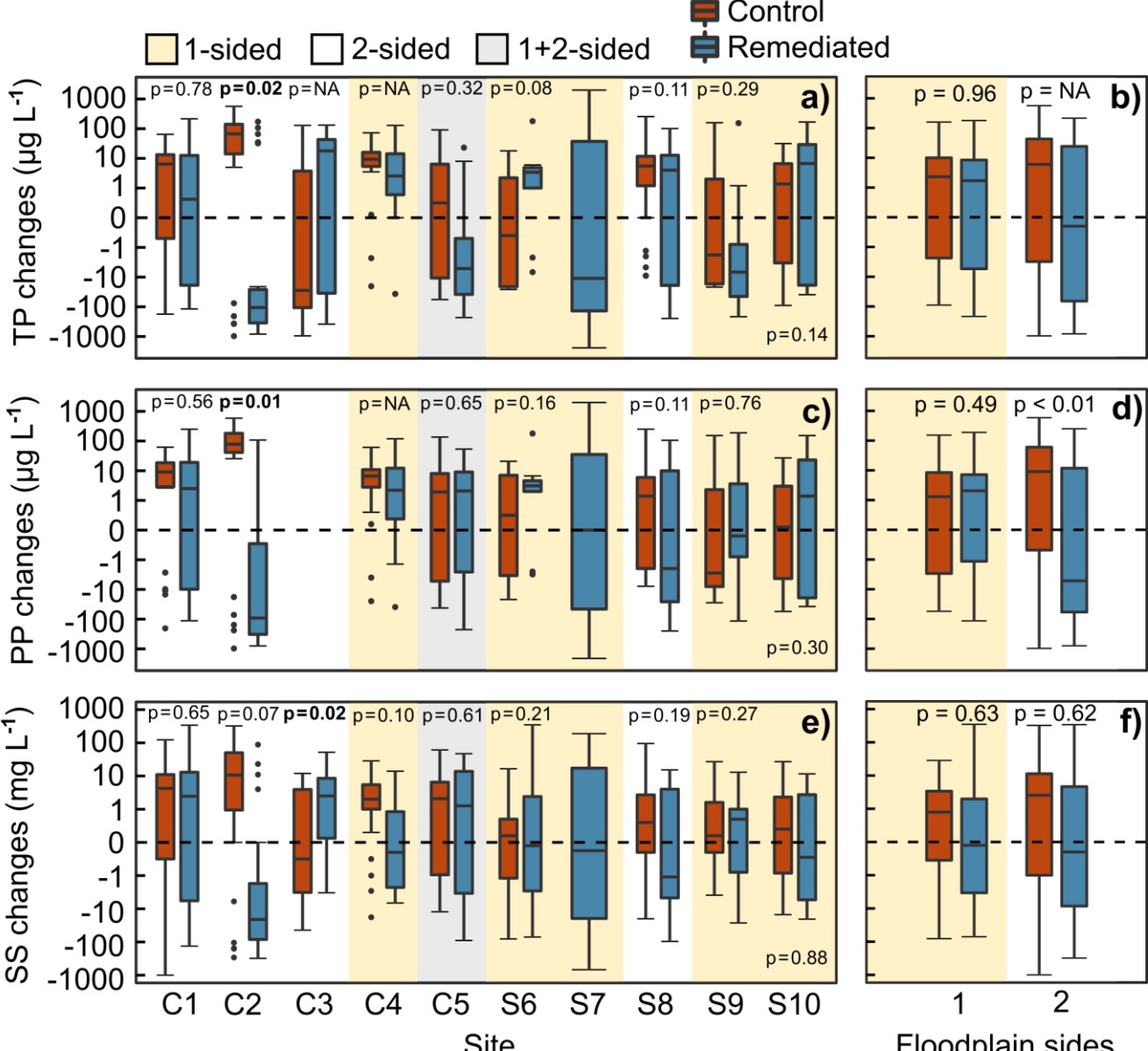

**Fig. 3.** Changes in TP, PP and SS concentrations (note log10 y-axis) between start and end of control streams (red boxes) and start and end of remediated streams (blue boxes) from April 2020 to June 2022. Changes in concentrations are shown a, c, e) by each site (C1–5 and S6–10) and b, d, f) sites grouped by number of floodplain sides along inset channel. When comparing floodplain sides, two sites were excluded due to lack of control stream (S7) and a combination of one- and two-sided floodplains along the reach (C5). P-values of permuted unpaired t-tests are shown within the panels. P-values of parametric and, when necessary, permuted paired t-tests are shown within the panels. P values = NA refers to non-normal and autocorrelated data that prevent reliable computation of t-tests. Bold font denotes significant difference (p < 0.05) between locations.

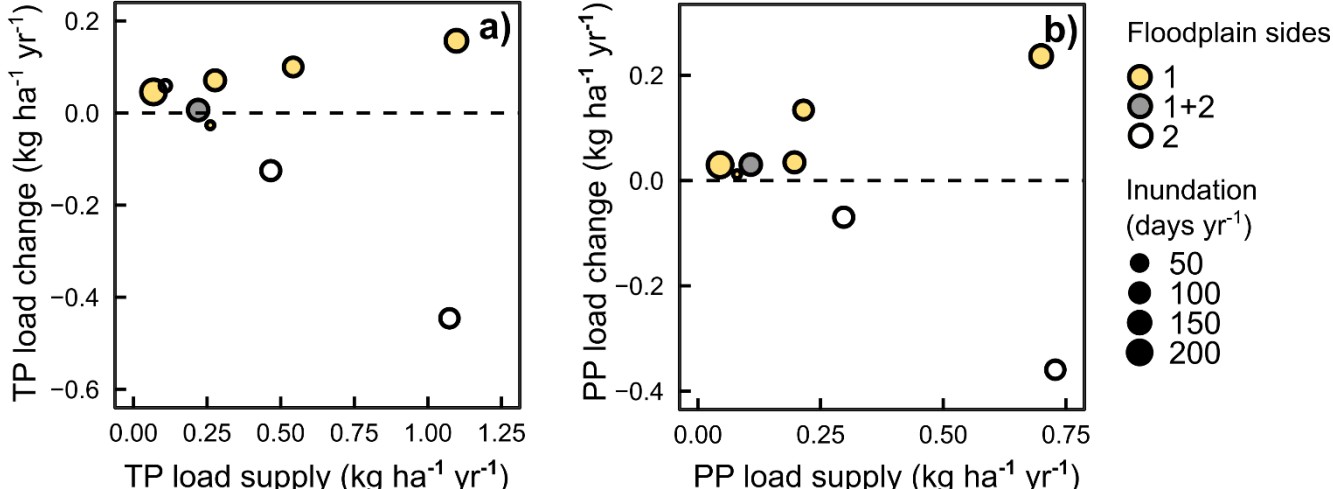

Fig. 4. Changes in a) TP and b) PP loads along remediated streams compared with P load supply loads at start of remediated streams. Remediated streams grouped as one-sided (yellow circles), one- and two-sided (gray circles) and two-sided floodplains (white circles). Size of circles denote average floodplain inundation days yr$^{-1}$. TP and PP loads of site C2 were excluded due to lack of flow data at downstream location and PP loads in site C3 was excluded since this was not measured.


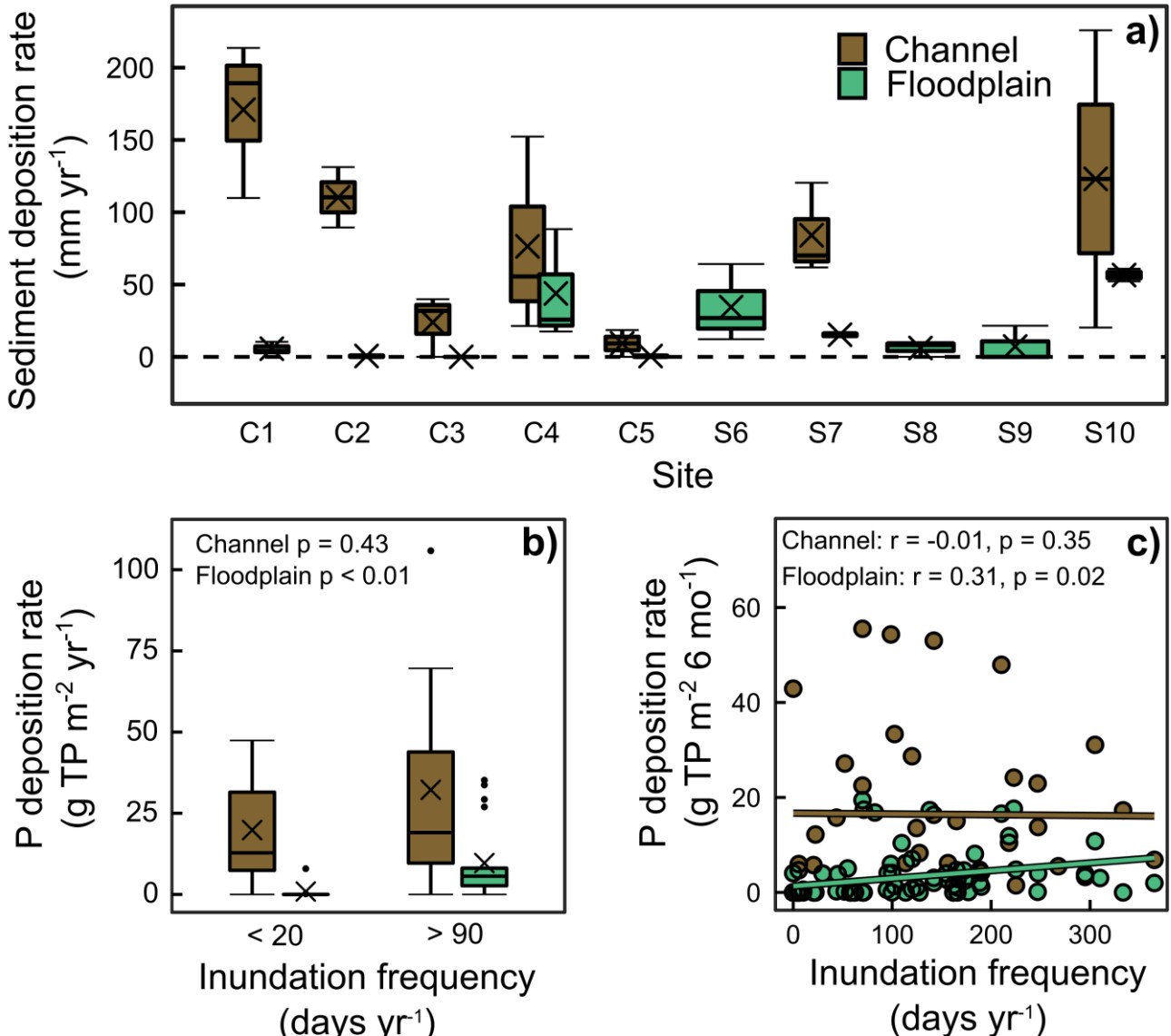

Fig. 5. a) Sediment deposition rate in depth on channel beds and floodplains across sites (C1–5 and S6–10) from sediment plates deployed between September 2021 and May 2022. b) Annual P deposition rate grouped by floodplain inundation frequency classes and c) biannual P deposition rate regressed against inundation frequency during each 6 month sampling occasion. P-values of permuted unpaired t-tests and permuted Pearson correlations are shown within panels.

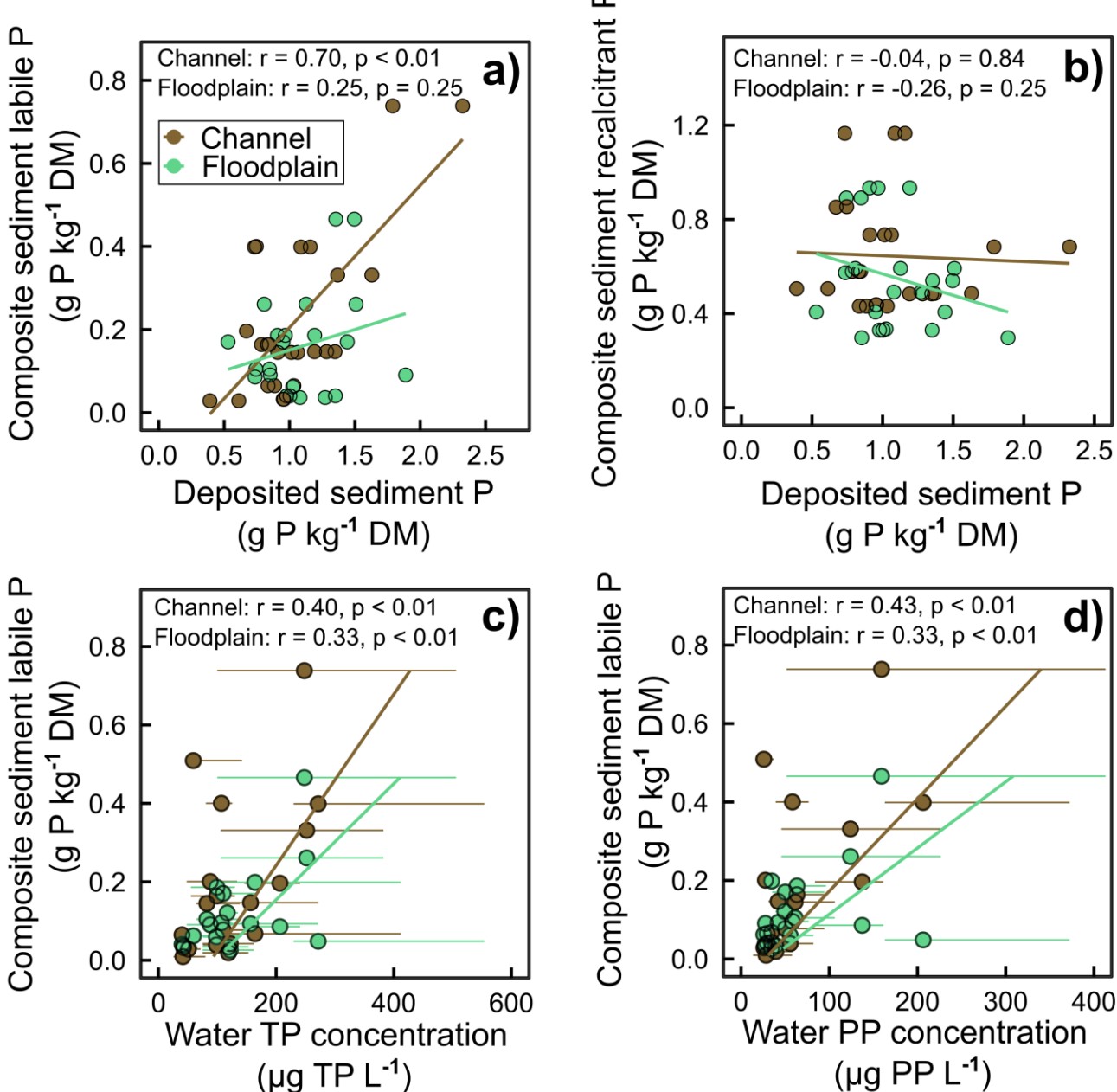

**Fig. 6.** Linear regressions between deposited sediment TP and composite sediment a) labile P and b) recalcitrant P in channel sediments (blue circles) and floodplain sediments (red circles). Linear regressions of composite sediment labile P by c) stream water TP concentrations and d) stream water PP concentrations. Deposited sediments were collected between September 2021 to May 2022, composite sediments sampled in spring 2022 and monthly water grab samples between April 2020 to July 2022. Zero values from plates without deposited sediments were removed from analysis. Labile P represent P fractions $H_2O$-P and Fe-P. Pearson r and p-values are shown within each panel and error bars show standard deviations of stream water P.