# Peer review of "Phosphorus supply and floodplain design govern phosphorus reduction capacity in remediated agricultural streams"

_EGUsphere, 2023_

## Author Response (AR1)

**Authors' response revision 2**

| RC1 comments | Authors' response |
|---|---|
| | (Line numbers refer to submitted pre-print version) |
| Title: highlight the main conclusion of the paper instead of a question? | Title was changed to: "Phosphorus supply and floodplain design govern phosphorus reduction capacity in remediated agricultural streams." |
| P concentration and loads are estimated with monthly data, which is a very low sampling frequency for phosphorus. Please provide data on load estimation uncertainty in similar context in Sweden (I know high frequency data exist in some catchments) and discuss how big these uncertainties are in comparison with the difference in loads between the upstream and the downstream sampling location. | At a site 1 km downstream of site C3, we used existing 'accurate' loads from the national monitoring programme (calculated from fortnightly flow-proportional sampling) as benchmark for validating flow-weighted (FW) loads. At the same location, we used discrete concentration samples of TP, PP and SS for FW load estimation. TP and SS were sampled 3 times each month and PP was sampled 2 times each month. From these, we constructed 2-3 variations of monthly concentrations, sourced from different days within each month. 2-3 variations of annual loads were subsequently estimated from the series using FW method. These were tested against accurate loads by calculating RMSE and % underestimation. Loads were validated for the same time period as the study.

Details on additional water samples were added: "Existing TP, PP and SS water chemistry from a location 1 km downstream of site C3 (Kyllmar et al., 2014) were used for subsequent validation of flow-weighted mean concentration (FW) load estimation method. Two types of data were used: 1) Annual loads estimated from fortnightly flow-proportional composite sampling and 2) fortnightly discrete concentration samples.", line 158, p 6.

Load validation was explained as: "The accuracy of annual load estimation was tested downstream of site C3. Here, loads calculated from existing fortnightly flow-proportional composite sampling represent 'accurate' loads (Dialameh & Ghane, 2022). At the same location, FW loads were calculated for the study period using water samples from two to three different days within each month to compare with true loads (Fig. S11). At this location, annual FW loads were underestimated across all parameters (TP: -28 %, PP: -29 % and SS: -52 %).", line 169, p 6.

Load validation was discussed as: "Further, flow-weighted average concentration load estimation is generally conservative (Elwan et al., 2018), which also was shown in proximity of site C3. Therefore, the annual TP and PP loads may in fact be underestimated with 30 %.", line 248, p 8. |

| | |
|---|---|
| Is the term "floodplain" correct to refer to the flat section of a two-stage ditch? Please provide some definition of reference to papers that have used the same term. | In the context of agricultural stream/ditch remediation, the term floodplain has been used regularly to denote the excavated 2-stage section.

Definition of floodplains added: "Here, floodplains refer to 2 to 6 m wide benches that are inundated upon higher flows (D'Ambrosio et al., 2015; Mahl et al., 2015; Powell et al., 2007)", line 55, p 2. |
| It would be interesting to compare the results of the engineering techniques used in the paper with river restoration approaches (including re meandering). What are the pros and cons? How does P retention compare? | Additional discussion on re-meandering was added: "Although the combination of re-meandering and floodplains resulted in lower SS and P export in site S8, stable meanders can be difficult to establish without additional protective measures to avoid erosion and migration (Wohl et al., 2015). Results on P mitigation with re-meandered channels are scarce but it has been shown to improve stream habitat quality (Lorenz et al., 2009). However, meanders rely on a certain freedom of channel migration that can infringe on productive agricultural land (thus less frequently implemented) and also lead to higher sediment (and thus P) exports and are therefore less suitable in narrow riparian buffers with limited room for channel evolution.", line 369, p 12.

P reductions in site C2 were compared with another study that reported P reductions from floodplains and low-grade weirs. Following sentence was added: "Littlejohn et al. 2014 reported 45 % TP load reductions with two low-grade weirs in streams with constructed floodplains, comparable to that of site C2.", line 365, p 12. |
| I would have expected a longer paragraph about the long-term management of those ditches. Can we expect different results with different management technique (dredging frequency, grass cutting) for the same geometry? | Added discussion on vegetation removal: "Dense vegetation on floodplains of remediated streams also offer opportunities for removing P accumulated in biomass, as dry floodplains are more accessible for mowing and collection compared to submerged macrophytes in channel. However, woody vegetation on banks and floodplains can physically obstruct operations to remove biomass.", line 378, p 12.

One aim with remediated streams is to reduce the need for costly routine dredging. Dredging also negatively affects stream biota and ecosystem functions and we do therefore not recommend increased dredging frequency, as already stated on lines 369-378, p 12.

Other studies have confirmed that remediated streams reduce channel sedimentation, but in our study sedimentation is still high in channels. Despite this, we argue that the room for allowing |

| | sedimentation and thus P retention is higher in remediated streams, which can reduce the need for dredging. |
|---|---|
| Lateral inputs are ignored in this paper. Can we neglect them, or suppose they are similar in the control reaches and the two-stage reaches? | Lateral inputs in these systems are dominated by tile drain and groundwater input along both control and remediated streams (as shown by high BFI that do not differ between control and remediated reaches). Since both stream types share the same land use (as shown in Fig. S1-5) and similar tile drainage systems, we assume that lateral P inputs are comparable between the two.

Sentence in discussion was added: "We assumed that lateral inputs (tile drains and groundwater inflow) along both remediated and control reaches were comparable and thus not influencing the comparison between the two reaches. All paired reaches received tile drain inputs from adjacent fields with identical crop cultivation and flat topography with predominant subsurface flow pathways. There were no large deviations in loads between up- and downstream and we do therefore not have any reasons for suspecting significant lateral hotspots.", line 262, p 9. |
| Fig 3 and throughout the manuscript: show the total filtered P data (instead or in addition to PP, which is estimated by difference) | The main question of the paper is if remediated streams reduce the particulate fraction of P, due to either sedimentation or erosion control. PP is therefore the fraction of interest (along with SS) that can be deposited and/or exported from mass erosion. Filtered TP can be estimated from this data, since both TP and PP are shown. |
| L40 "and increasingly so amidst a changing a climate with accelerated pollution" consider rephrasing | Sentence rephrased to "increasingly so with a changing a climate that accelerates pollution to freshwaters", line 40, p 2. |
| L43 "Without recurring and costly maintenance, internal erosion from trapezoidal channels can therefore contribute with higher sediment and P loads compared to distal sediment sources in catchments" add reference | Maintenance, i.e., dredging does not reduce erosion, it rather maintains the fluvial disequilibrium. Therefore, we removed: "Without recurring and costly maintenance" and changed sentence to: "Internal erosion from trapezoidal channels can therefore contribute with higher sediment and P loads compared to distal sediment sources in catchments (Simon and Rinaldi, 2006).", lines 43-45, p2. |
| L85 "Site C1 two tributaries draining to remediated stream reach" consider rephrasing | Sentence rephrased to "Site C1 has two tributaries draining into the remediated stream.", lines 85-86, p3. |
| section 2. please provide a figure or photo of the sediment plates | Photographs of sedimentation plates were added in new Fig. S10. |
| L192: please provide correlation coefficient for each site, not only a global coefficient of correlation because the SS-PP relationship may vary among sites. | Correlation coefficients added in new Table S3. Sentence was changed to: "Concentrations of SS ranged between < 1 to 1200 mg SS L-1 and were correlated to PP concentrations at each site (Table S3), but to a lesser degree in site C1 and S9 at high concentrations.", line 192, 7. |

| | |
|---|---|
| L204 and fig 4 true but this conclusion is based on a very limited number of sites, and there is no overall pattern when looking at the two-sided and one-sided ditches together. Please discuss. Isn't the opposite pattern for the two types of ditches suspicious? Wouldn't we expect the one-side ditches to lay in-between the two-side ditches and the control? | New sentence was added in discussion: "However, P load data were only available for two sites with two-sided floodplains which limits the certainty of this floodplain design being the main control responsible for P load reductions, warranting further study of floodplain designs.", line 248, p 8.

The lack of overall pattern we argue is due to the inefficiency of the studied 1-sided floodplains. Our explanation for this is already mentioned: "In one-sided designs, banks next to inset channels were also exposed to higher stream velocities; the greater water depth in channels compared to floodplains led to higher water velocities that exerted higher shear stress on banks adjacent to the channel.", lines 269-271, p 9.

It is true that not all 2-sided were reducing P, but none of the 1-sided showed any improvements in comparison. It is still possible that 1-sided floodplains can be successful in certain circumstances. For example, site S7 has good design properties and placement, but we lack a representative reference at this site needed to validate its efficacy.

We also expect 1-sided to perform better than control streams. However, there is no load data upstream of control reaches and this prevents us from calculating load changes along these reaches for comparison. |
| L235: "Total P in newly deposited sediments on plates was 1.6 times higher than in composite sediments…" I found this paragraph difficult to understand. The difference between newly deposited sediments and composite sediments, and how to interpret this difference, was not clear to me. This could be improved in the Materials and methods section 2.2 and 2.3 | Distinction between newly deposited and composite sediments is clarified as "In addition to newly deposited sediments on plates, composite sediments that represent the integration of both short- and long-term sediment deposition were sampled with a trowel in spring 2022 to determine P fractions.", lines 118-119, p 4.

Additional composite sediment definition was added in results: "composite sediments, i.e., sediments down to 5 cm depth outside of plates.", line 235, 8. |
| L249 "it is recommended to…" / "our results demonstrate that…" I would not make strong recommendation based on those results. Replace with more cautious phrasing like "the data suggests that…" | First sentence was rephrased to: "to realize their P reduction potential, the results indicate that remediated streams should be targeted in areas with P loading exceeding 0.25", lines 249-250, p 9.

Second sentence was rephrased to: "Our results suggest that PP reductions can be…", line 252, p 9. |
| L251 "corresponding to a considerable share of agricultural land within the contributing catchment." How much is considerable? (or delete this part of the sentence) | The following part was deleted: "corresponding to a considerable share of agricultural land within the contributing catchment.", lines 250-251, p 9. |

| | |
|---|---|
| L260 "equal proportions of PP and reactive P" is it reactive P or total filtered P? | It refers to soluble reactive P. All mentions of reactive P in manuscript were changed to SRP. |
| L355 "enhance P reductions" replace with "retention" | Rephrased as "this design is insufficient to reduce P concentrations and loads compared to control streams" to keep terminology consistent. Lines 354-355, 12. |

**RC2 comments**                                    **Authors' response**

| The data analysis and discussion of the timing of P retention should be expanded. A major conclusion of the paper is that P retention in 2-sided floodplain streams occurs during inundation events, but not during baseflow (Lines 201-203). However, it isn't clear how this was calculated. Was this based on stream concentration data collected during both baseflow and inundation events? I recommend showing this comparison in a figure for both remediated stream types and the controls. Does this difference also explain how P concentration can increase across the remediated reach, but load decreases (see site C1 in Table S2)? Two of the four 2-sided floodplain sites have average increases in PP concentration. What does this say about the effectiveness of these sites for removing PP? | Changes in P load during baseflow/inundation were calculated using monthly concentration data, resulting in g P day$^{-1}$. Calculation was explained in: "Daily loads of TP based on monthly water chemistry were also calculated by multiplying with daily flow", line 169, p 6.

S8 was the only site with significantly higher TP load reduction during inundation. No other 2-sided site showed this effect and we have therefore removed the comparison of 2-sided vs. 1-sided for temporal load reduction. Sentence was changed to: "Based on monthly water chemistry data, TP loads (g day-1 ha-1) were reduced during inundation events in site S8, compared to base flows (permuted t-test, p = 0.02). This effect was not observed at any other site.", lines 201-203, p 7.

Differences in load and concentration changes in C1 is not based on reductions during inundation. This is due to load normalization to ha draining area. Downstream kg P ha-1 yr-1 is lower since it drains a larger area. There is no difference between up- and downstream for kg P yr-1.

See explanation below (3rd comment) about 2-sided efficacy. |
|---|---|
| The average load and concentration changes in the remediated streams (lines 199-200) are based on a small sample size. For example, there are only two 2-sided floodplain streams with PP loads, and only 3 with TP loads. First, why weren't loads calculated for C2 and C3? Second, please note in the text this small sample size and how that impacts the results. You make strong claims about the effectiveness of these remediations but should mention the limited number of sites. Finally, the text has 0.22 kg PP/ha/yr for 2-sided floodplains but the abstract has 0.21. | Loads were not calculated in C2 at downstream of remediated reach since no reliable stage-rating curve could be established and thus no flow data. In C3, filtered TP was not measured, preventing PP calculation. This is clarified in figure text of Fig. 4: "TP and PP loads of site C2 were excluded due to lack of flow data at downstream location and PP loads in site C3 was excluded since this was not measured."

Added discussion on small sample size: "However, P load data were only available for two sites with two-sided floodplains which limits the certainty of this floodplain design being the main control responsible for P load reductions, warranting further study of floodplain designs.", line 248, p 8.

Sentence was changed to less strong phrasing on implications: "to realize their P reduction potential, the results indicate that remediated streams should be targeted in areas with P loading exceeding 0.25", lines 249-250, p 9.

Second sentence was changed to: "Our results suggest that PP reductions can be…", line 252, p 9. |

| | Load was corrected to 0.21 kg PP/ha/yr on line 200, p 7. |
|---|---|
| A similar concern to above, but how much of the benefits you see of 2-sided floodplains is the result of data from C2? This site has by far the greatest reduction in P and SS concentrations, which you attribute to the fact there is a perched culvert at the outlet which encourages settling. If you remove this site from your analysis, you may see little effect of 2-floodplain streams. | Site C2 account for most of the TP and PP reduction in concentrations. When removed from analysis, unpaired t-test of PP concentration changes in 2-sided floodplains are not significant (p = 0.22).

Among 2-sided sites, only C2 and S8 reduced TP, PP and SS concentrations, and not C1 and C3. This is clarified with addition in discussion: "However, among two-sided floodplain sites, C1 and C3 did not reduce P and SS, implying that solely implementing two-sided floodplains is insufficient and that further requirements such as frequent inundation, floodplain stability and reach lengths must be met by proper design of two-sided sites", line 248, p 8. |
| Why didn't you use a paired t-test? These are paired samples (upstream control and downstream restored). A paired t-test would compare calculated concentration differences from the same collection day and site. This would be an interesting additional comparison that could give additional insight beyond the lumped comparison currently in the paper. | Fig. 3 was amended with p-values of paired t-tests by each site, with matching dates for up- and downstream. Sites with non-normal concentration distributions were tested with permuted paired t-tests, described in the figure text in Fig. 3.

Sentence was added in results: "Concentration reductions in TP and PP among sites with two-sided floodplains were mostly explained by site C2 but also S8 (Fig. 3).", line 201, p 7. |
| Lines 36-37: Note that these channels have been artificially straightened, deepened, and shaped into prismatic channels. | Sentence was changed to: "comprise of artificially straightened and deepened trapezoidal channels", line 36, p 2. |
| Lines 85-86: "Site C1 two…" this sentence is incomplete. | Sentence was changed to: "Site C1 has two tributaries draining into the remediated stream", lines 85-86, p 3 |
| Lines 115-116: Should "and" be "over" ("over 6 months…")? | Sentence was changed to: "deposition rates per area over six months ", line 115, p 4. |
| Line 148: Slope is dimensionless in the stream power equation (e.g. 1% slope should be 0.01). There may be in error in your unit stream power calculations (Table 1). The values seem exceptionally high, and you report slope as %. I recommend rechecking these calculations. Also please report the Q value (full inset channel flow) that you used in the calculation. | Sentence was changed to: "S is dimensionless channel bed slope" 147-148, p 5.

Unit stream power was re-calculated with dimensionless slope (previous was indeed with % slope). Unit stream power values were corrected in Table 1. Corrected unit stream power did not change the ordinations in Fig. S13 and its correlation with P sedimentation.

Channel max flow is now reported in Table 1. |
| Line 149: Should cite some of the extensive work on linking stream power to sediment transport capacity (e.g., Bagnold, Parker) | References to Bagnold and Reinfelds added to the sentence on lines 143-144, p 5. |

| | |
|---|---|
| Section 2.6: I appreciate you referring to the R packages you used. I suggest citing them as well to give the authors credit. | All non-base R packages were cited on lines 161-162, 174 and 185, p 6. |
| Lines 143-144: Did you also test whether the absolute TP concentrations were correlated with hydrologic metrics? | Yes, TP concentrations in any of the three sampling locations were uncorrelated to BFI and RBI and therefore not reported. |
| Lines 203-204: This final sentence is confusing, please revise. Higher incoming loads in two-sided floodplains led to greater load reduction, while higher loads in one-sided floodplains led to greater load increases along the remediated reaches. | Sentence was changed to: "Higher incoming loads in two-sided floodplains led to greater load reduction, while higher loads in one-sided floodplains led to greater load increases along the remediated reaches", line 203, p 7. |
| Lines 216-219: I don't see how these analyses support the claim that unit stream power alone during low flows is the primary driver for P deposition on sediment beds. | The calculated unit stream power when inset channel is full represents an indicator for the channel's ability to transport sediments. This stream power indicator correlated with P deposition, which TP loads did not.

This is explained in the existing section: "On channel beds, P deposition rates increased with lower unit stream power (r = -0.43, p = 0.03; Fig. S12), representing the maximum stream power per m-2 bed surface area when flow was confined to the inset channel. Deposited P did not correlate with the average of stream water TP loads for each 6 month sampling period (r = -0.16, p = 0.42; Fig. S12) and neither with the ratio of unit stream power:TP loads (r = 0.28, p = 0.32), indicating that unit stream power alone during lower flows was the primary driver for P sedimentation on channel beds.", lines 215-219, p 7-8. |
| Line 225: Please clarify "increased with a magnitude of 1". The figure makes it look like the difference in average deposition rates are > 1. | Average deposition on floodplains with > 90 inundation days was 10.7 times greater than floodplains with < 20 inundation days = 1 order of magnitude.

Sentence was changed to: " P deposition rates on floodplains were 10.7 times higher during > 90 inundation days", line 225, p 8. |
| Lines 378-379: I suggest revising or removing this final sentence. You should provide citations that P exports will increase (as will flood and drought frequency) with climate change. Also, it is unclear how stream remediation will help with flood and drought control (this is the first place drought is mentioned). | The term drought control was removed and sentence was changed to: "As future climate will bring higher frequencies of hydrological extremes and consequently higher P exports (Mehdi et al., 2015; Ockenden et al., 2017), remediation of agricultural streams can offer improved flood control together with enhanced water purification.", lines 378-379, p 12.

The greater bankfull capacity of remediated streams (ca 3x volume) buffers flooding on adjacent fields during high flows. |
| Line 385: Again, there is no other mention of how 2-stage ditches help with droughts. I suggest removing this. | The term drought was removed from this sentence, line 385, p 13. |

---

## Author Response (AR2)

**Authors' response revision 3**

Dear Editor,
In response to the addressed inconsistencies in the discussion, we have clarified the influence of site C2 on the effect of P reductions among the two-sided floodplain group and stated that the effect of C2 also is influenced by its perched culvert (lines 259-262, p 9). The sentence: "Our results suggest that PP reductions can be substantial in these systems and lead to decreased TP loads, adding to previous body of research from the US" was removed (lines 272-273, p 9).

Headline 4.1 was changed to "Influence of floodplain designs on P reductions in stream" to reflect the limited support for two-sided designs as the most important control of P reductions.

Sentence on load underestimation in discussion was clarified as "Further, flow-weighted average concentration load estimation from monthly water samples is generally conservative (Elwan et al., 2018), which also was the case in proximity of site C3 where loads based on monthly samples were 30 % lower than loads from biweekly flow-proportional sampling." (lines 266-268, p 9).

We also revised the phrasing of inundation effect on TP loads in discussion (lines 295-298, p 10; 393-395, p 13). TP loads were only significantly reduced during inundation events compared to base flows in site S8 and not across all sites as previously stated.

In the conclusion, we changed phrasing to "P reductions can be achieved with two-sided floodplains" instead of "floodplain designs controlling P reductions" (lines 435-438, p 14).

Line and page numbers refer to manuscript with track changes.